# A secreted LysM effector protects fungal hyphae through chitin-dependent homodimer polymerization

Andrea Sánchez-Vallet[1,2], Hui Tian[1], Luis Rodriguez-Moreno[1,¤a], Dirk-Jan Valkenburg[1], Raspudin Saleem-Batcha[3,¤b], Stephan Wawra[4], Anja Kombrink[1], Leonie Verhage[1], Ronnie de Jonge[1,¤c], H. Peter van Esse[1,¤d], Alga Zuccaro[4], Daniel Croll[5], Jeroen R. Mesters[3‡*], Bart P. H. J. Thomma[1,4‡*]

1 Laboratory of Phytopathology, Wageningen University& Research, Wageningen, The Netherlands, 2 Plant Pathology Group, Institute of Integrative Biology, ETH Zurich, Zurich, Switzerland, 3 Institute of Biochemistry, Center for Structural and Cell Biology in Medicine, University of Lübeck, Lübeck, Germany, 4 University of Cologne, Institute for Plant Sciences, Cluster of Excellence on Plant Sciences (CEPLAS), Cologne, Germany, 5 Institute of Biology, University of Neuchâtel, Neuchâtel, Switzerland

☯ These authors contributed equally to this work.
¤a Current address: Departamento de Biología Celular, Genética y Fisiología, Universidad de Málaga, Málaga, Spain.
¤b Current address: Institute of Pharmaceutical Sciences, University of Freiburg, Freiburg, Germany
¤c Current address: Plant-Microbe Interactions, Department of Biology, Science4Life, Utrecht University, Utrecht, The Netherlands
¤d Current address: The Sainsbury Laboratory, University of East Anglia, Norwich Research Park, United Kingdom
‡ These authors also contributed equally to this work.
* jeroen.mesters@uni-luebeck.de (JRM); bart.thomma@wur.nl (BPHJT)

**Data Availability Statement:** All whole-genome sequencing data is accessible on the Nucleotide Short Read Archive (accession numbers

## Abstract

Plants trigger immune responses upon recognition of fungal cell wall chitin, followed by the release of various antimicrobials, including chitinase enzymes that hydrolyze chitin. In turn, many fungal pathogens secrete LysM effectors that prevent chitin recognition by the host through scavenging of chitin oligomers. We previously showed that intrachain LysM dimerization of the *Cladosporium fulvum* effector Ecp6 confers an ultrahigh-affinity binding groove that competitively sequesters chitin oligomers from host immune receptors. Additionally, particular LysM effectors are found to protect fungal hyphae against chitinase hydrolysis during host colonization. However, the molecular basis for the protection of fungal cell walls against hydrolysis remained unclear. Here, we determined a crystal structure of the single LysM domain-containing effector Mg1LysM of the wheat pathogen *Zymoseptoria tritici* and reveal that Mg1LysM is involved in the formation of two kinds of dimers; a chitin-dependent dimer as well as a chitin-independent homodimer. In this manner, Mg1LysM gains the capacity to form a supramolecular structure by chitin-induced oligomerization of chitin-independent Mg1LysM homodimers, a property that confers protection to fungal cell walls against host chitinases.

PRJNA327615 and PRJNA178194). The atomic coordinates and experimental structure factors were deposited with the Protein Data Bank under accession code 6Q40.

**Funding:** Work in the laboratory of B.P.H.J.T. is supported by the Research Council Earth and Life Sciences (ALW) of the Netherlands Organization of Scientific Research (NWO). Part of the work was funded by the Deutsche Forschungsgemeinschaft (DFG, German Research Foundation) under Germany´s Excellence Strategy – EXC 2048/1 – Project ID: 390686111. The funders had no role in study design, data collection and analysis, decision to publish, or preparation of the manuscript.

**Competing interests:** The authors declare no conflict of interest exists.

## Author summary

Chitin plays a central role in plant-fungi interactions, since it is a major component of the fungal cell wall that is targeted by host hydrolytic enzymes to inhibit the growth of fungal pathogens on the one hand, and release chitin fragments that are recognized by host immune receptors to activate further immune responses on the other hand. In turn, many fungal pathogens secrete chitin binding LysM effectors to which currently two functions have been assigned. Most LysM effectors that were functionally characterized to date function to prevent chitin recognition by host immune receptors through chitin sequestration. Additionally, some LysM effectors were shown to protect fungal hyphae against hydrolysis by host chitinases. The crystal structure of Mg1LysM from the Septoria blotch pathogen of wheat, *Zymoseptoria tritici*, revealed that chitin-induced dimerization of two Mg1LysM protomers through high affinity binding is required for hyphal protection against chitinases. Since Mg1LysM also forms ligand-independent homodimers, a supramolecular structure can be formed in which chitin-induced oligomerization of Mg1LysM ligand-independent homodimers form a contiguous Mg1LysM higher ordered structure that is anchored to the chitin in the fungal cell wall to prevent hydrolysis by host chitinases.

## Introduction

Fungi constitute an evolutionarily and ecologically diverse group of microorganisms that are characterized by the presence of chitin, an *N*-acetyl-D-glucosamine (GlcNAc) homopolymer, in their cell walls. In addition to providing strength, shape, rigidity and protection against environmental hazards, chitin is also a well-known inducer of plant immune responses [1,2,3]. A major mechanism of plant defense against fungal invasion includes the secretion of microbial cell wall-degrading enzymes that include chitin-degrading enzymes, known as chitinases, to hinder fungal pathogen ingress [4,5]. Plant chitinases are diverse in nature, and grouped into six different classes that belong to glycosyl hydrolase families 18 and 19 [6,7]. Although many chitinases are specifically produced upon pathogen invasion, others are constitutively expressed [4]

Chitin hydrolysis on the one hand inhibits fungal growth, and on the other hand releases chitooligosacharides [8,9](Kasprzewska, 2003; Liu et al, 2014) that are recognized by cell surface receptors of host cells to mount an appropriate immune response [3,10]. In plants, chitin is recognized in the extracellular space through membrane-exposed Lysin motif (LysM)-containing receptor molecules [1,11,12,13]. In turn, many successful fungal pathogens have evolved effector molecules that either protect their cell walls against plant chitinases or prevent or perturb the elicitation of chitin-triggered host immunity [3,10,14,15,16].

Since decades, the interaction between the foliar fungal pathogen *Cladosporium fulvum* and its only host tomato has been studied to unravel the role of pathogen virulence and host defense mechanisms, including mechanisms that revolve around chitin [17]. After leaf penetration, *C. fulvum* secretes an arsenal of apoplastic effector proteins, including the chitin-binding effector proteins Avr4, which protects fungal hyphae against hydrolysis by plant chitinases [18,19], and Ecp6, which perturbs the activation of chitin-triggered host immunity [14,20]. Whereas Avr4 binds chitin through an invertebrate chitin-binding domain [21,22], Ecp6 utilizes LysM domains for chitin binding [14,20]. Previous biochemical analysis revealed that Avr4 monomers require a stretch of at least three exposed GlcNAc residues for binding, and positive allosteric interactions among Avr4 molecules occur during chitin binding to facilitate the shielding of cell wall chitin against host chitinases [21]. Based on X-ray crystallography it

was recently shown that two Avr4 molecules interact through their chitohexaose ligand to form a three-dimensional molecular sandwich that encapsulates two chitohexaose molecules within the dimeric assembly [23]. A crystal structure of Ecp6 revealed chitin-induced dimerization of two of the three LysM domains, resulting in the formation of an ultrahigh affinity (pM) chitin-binding groove, conferring the capacity to outcompete plant receptors for chitin binding [20]. Interestingly, whereas Avr4 homologs are found in other, *C. fulvum*-related, Dothideomycete plant pathogens [24], LysM effectors are widespread in the fungal kingdom [25,26,27]. In several plant pathogenic fungi, including the Dothideomycete *Zymoseptoria tritici* and the Sodariomycetes *Magnaporthe oryzae*, *Colletotrichum higginsianum* and *Verticillium dahliae*, LysM effectors have been shown to contribute to virulence through chitin binding [15,28,29,30].

The LysM effectors Mg1LysM and Mg3LysM, with one and three LysM domains, respectively, have been characterized from the Septoria tritici blotch pathogen of wheat, *Z. tritici* [15]. Functional characterization has revealed that Mg3LysM can suppress chitin-induced immunity in a similar fashion as *C. fulvum* Ecp6. Surprisingly, in contrast to Ecp6 and similar to Avr4, Mg3LysM was additionally shown to have the ability to protect fungal hyphae against chitinase hydrolysis. As expected, based on the presence of a single LysM domain only, a role in suppression of chitin-triggered immunity could not be demonstrated for Mg1LysM [15]. Intriguingly, however, Mg1LysM was characterized as a functional homolog of Avr4 that protects hyphae against hydrolysis by host chitinases [15]. In order to understand how a LysM effector that is composed from little more than only a single LysM domain is able to confer protection of cell wall chitin from hydrolysis by plant enzymes, we aimed to obtain a crystal structure of the *Z. tritici* effector Mg1LysM in this study. Surprisingly, we discovered that Mg1LysM has the ability to simultaneously undergo ligand-mediated dimerization as well as ligand-independent homodimerization, allowing the formation of a contiguous oligomeric structure that anchors to the fungal cell wall through chitin to confer its protection ability.

## Results

### Crystal structure of Mg1LysM reveals ligand-dependent and -independent intermolecular dimerization

In order to understand LysM effector functionality, and particularly how Mg1LysM is able to protect chitin against chitinase hydrolysis, a crystal structure of Mg1LysM was determined. To this end, Mg1LysM was heterologously produced in the yeast *Pichia pastoris* and purified based on the presence of a His-FLAG affinity tag. The large Mg1LysM crystals that were finally obtained by micro-seeding techniques [31] belonged to the space group $P\,6_1\,2\,2$. Some crystals were soaked with the $Ta_6Br_{14}$ cluster and initial phases were determined by the single-wavelength anomalous dispersion technique (SAD; Table 1). The initial phases were further improved with the help of an I3C soaked crystal. A native dataset was finally refined to a resolution of 2.41 Å with an Rwork and Rfree of 17.96% and 22.03%, respectively (Table 1). The structure model comprises in total four copies of the complete mature protein sequence except for the first amino acid after the signal peptide for chains A to C and the first two amino acids after the signal peptide for chain D, and one carbohydrate molecule per asymmetric unit (a. u.).

As expected, the tertiary structure of the LysM domain of an Mg1LysM monomer is similar to that of previously described LysM domains [13,20,32,33,34,35] with a conserved βααβ-fold in which the antiparallel β-sheet lies adjacent to two α-helices (Fig 1 and S1 Fig). The compact LysM structure is stabilized by two disulfide bridges between Cys[44] and Cys[78], and between

**Table 1. Data collection and refinement statistics.**

| | Native | Derivative-1 (SAD) $(Ta_6Br_{12})^{2+}$ | Derivative-2 (SIRAS) (I3C) |
|---|---|---|---|
| **Data collection statistics** | | | |
| Beamline | BL14.1—BESSY | | |
| Wavelength (Å) | 0.91841 | 1.24845 | 1.88313 |
| Space group | P $6_1$ 2 2 | | |
| Cell dimensions a, b, c (Å) | 119.4, 119.4, 157.7 | 119.5, 119.5, 157.6 | 119.3, 119.3, 157.3 |
| Resolution (Å) | 49.12–2.41 (2.54–2.41) | 26.97–2.96 (3.12–2.96) | 39.37–5.00 (5.27–5.00) |
| $R_{sym}$ [#] (%) | 8.3 (50.4) | 7.8 (20.1) | 12.9 (20.7) |
| I / σI [§] | 18.0 (4.5) | 28.2 (11.5) | 17.6 (17.3) |
| Completeness (%) | 92.1 (100) | 91.5 (100) | 99.8 (100) |
| Redundancy | 10.7 (10.7) | 17.6 (16.3) | 19.1 (19.9) |
| Anomalous completeness (%) | - | 91.4 (100) | 100 (100) |
| Anomalous multiplicity | - | 9.5 (8.6) | 11.1 (11.0) |
| **Phasing statistics** | | | |
| Figure of Merit (FOM) | - | 0.57 | 0.36 |
| Map Skew | - | 1.40 | 0.03 |
| Correlation of local RMS density | - | 0.92 | 0.60 |
| Correlation Coefficient (CC) | - | 0.56 | 0.27 |
| **Refinement statistics** | | | |
| Resolution (Å) | 2.41 | - | - |
| No. of reflections (total / free) | 24,112 / 1,219 | - | - |
| $R_{work}$ / $R_{free}$ [†] (%) | 17.96 / 22.03 | - | - |
| No. of atoms / average B-factor | | | |
| Overall | 2,595 / 55.2 | - | - |
| Protein | 2,419 / 55.6 | - | - |
| Ligand | 43 / 53.7 | - | - |
| R.m.s. deviations bond lengths (Å) | 0.010 | - | - |
| R.m.s. deviations bond angles (°) | 1.195 | - | - |
| Ramachandran plot (% favored / % outliers) | 95.8 / 0.0 | - | - |
| MolProbity (clash- & overall score) | 6.42 / 1.65 | - | - |

The values in the parentheses refer to the highest resolution shell.

[#]$R_{sym} = (\Sigma |I_{hkl} - <I_{hkl}>|) / (\Sigma I_{hkl})$, where the average intensity $<I_{hkl}>$ is taken over all symmetry equivalent measurements and $I_{hkl}$ is the measured intensity for any given reflection.

[§]I/σI is the mean reflection intensity divided by the estimated error.

[†]$R_{work} = ||F_o| - |F_c|| / |F_o|$, where $F_o$ and $F_c$ are the observed and calculated structure factor amplitudes, respectively. $R_{free}$ is equivalent to $R_{work}$ but calculated for 5% of the reflections chosen at random and omitted from the refinement process.

Cys[13] and Cys[70] (Fig 1 and S1 Fig). In addition to the single LysM domain, Mg1LysM comprises a relatively long N-terminal tail that contains a short β-strand (Fig 1).

The four Mg1LysM monomers within the a.u. form two pairs of homodimers that are each very tightly packed. The large homodimerization interface that occurs between two monomers was calculated to be 1113 Å2 using PISA (Protein Interfaces, Surfaces and Assemblies; http://www.ebi.ac.uk/pdbe/prot_int/pistart.html) [36]and is stabilized by a total of 25 hydrogen bonds between residues of each of the monomers. In addition, the crystal structure revealed that the N-terminal 12 amino acid tails of the homodimer run anti-parallel and form a small

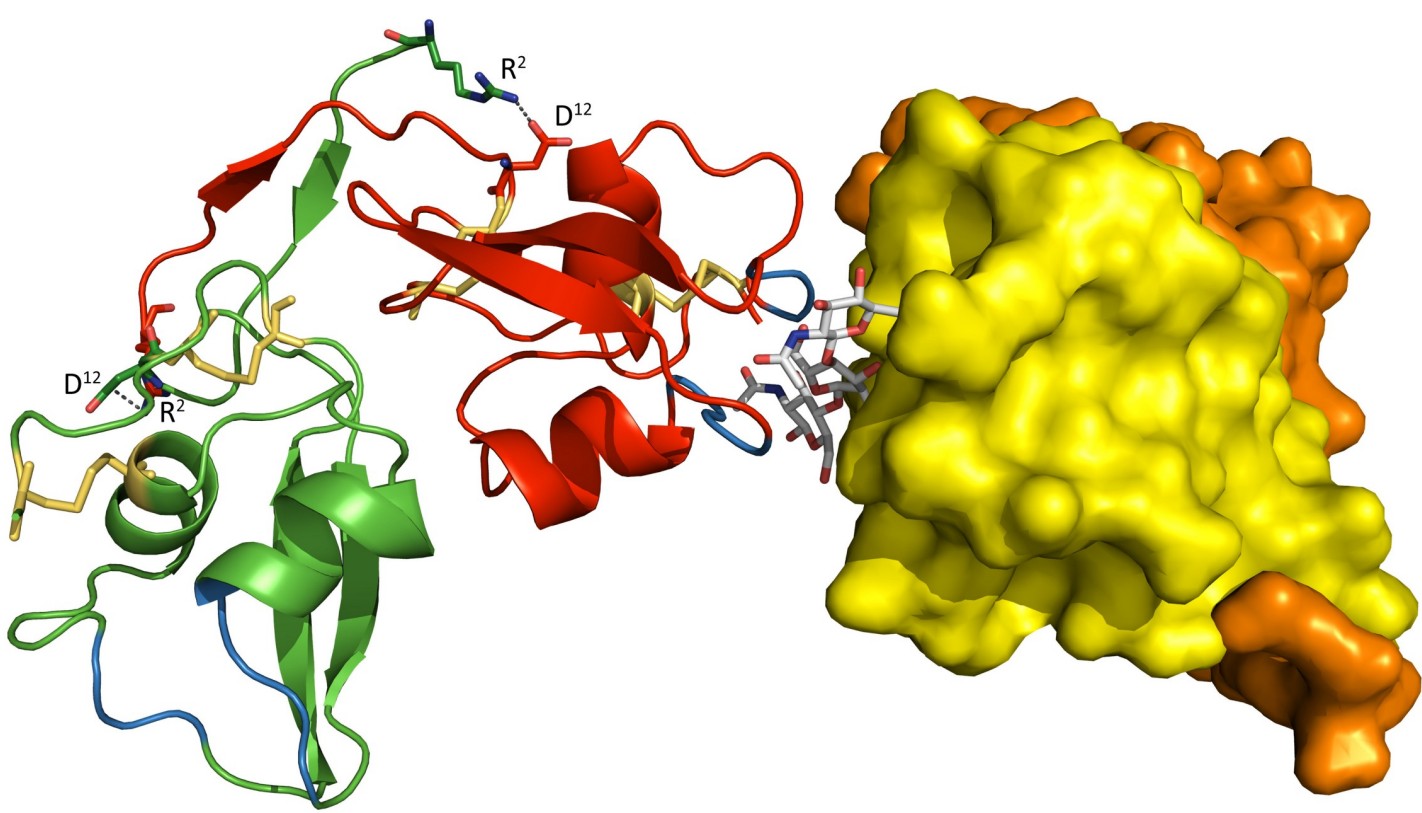

**Fig 1. Overall crystal structure of the *Zymoseptoria tritici* effector Mg1LysM.** Crystal structure model in which a dimer of two Mg1LysM homodimers is shown, with each of the Mg1LysM molecules in a different colour (orange, yellow, red and green). While the two monomers that form a ligand-independent homodimer on the right are represented as a surface model, the two monomers that form a ligand-independent homodimer on the left are represented by ribbons with the (putative) chitin binding sites indicated in blue and the disulfide bridges as yellow sticks. The chitin trimer that mediates the dimerization of two ligand-independent Mg1LysM homodimers is depicted by grey sticks. The two salt bridges between $R^2$ and $D^{12}$ in the dimer interface on the left are indicated with grey discontinuous lines.

but stable β-sheet ($^5$ITI$^7$ of each chain) that is stabilized by clustering of all four isoleucine side-chain residues and by threonine-threonine sidechain hydrogen bonding, potentially strengthening the homodimer (Fig 1). The latter hypothesis is further supported by the formation of two additional salt bridges formed between Arg$^2$ of one subunit and Asp$^{12}$ of the other one (Fig 1). The root mean square deviations (rmsd) between the Cα atoms of the two homodimers of the a.u. is 0.267 Å as calculated with Lsqkab of the CCP4 suite [37].

Surprisingly, when determining the crystal structure for the *C. fulvum* LysM effector Ecp6 in the absence of exogenously added chitin we found a chitin tetramer in a large interdomain groove between two of the three intrachain LysM domains that appeared to constitute an ultra-high affinity binding site, while no chitin binding was observed to the remaining, third LysM domain of Ecp6 [20]. Unexpectedly, the calculated 2|F$_0$|−|F$_c$| electron density map of the Mg1LysM crystal structure assembly similarly revealed well-defined electron density for a single chitin trimer bound to one monomer of the a.u. only (S2 Fig). Inspection of the crystal packing interactions unveiled the presence of a chitin binding pocket formed between two Mg1LysM protomers of neighbouring homodimers (Fig 1). Since protein purification and crystallization was performed without exogenous addition of chitin in this case as well, we concluded that the chitin once more was derived from the cell wall of the heterologous protein production host *P. pastoris*. Potentially, this finding indicates that Mg1LysM displays an increased affinity for chitin (low micromolar range) when compared with other, single-acting,

LysM domains. The chitin binding site is formed by the loops between the first ß-strand and the first α-helix and between the second α-helix and the second ß-strand of Mg1LysM, encompassing the residues $^{26}$GDTLT$^{30}$ and $^{56}$NRI$^{58}$ that are conserved in many other LysM domains including those of Ecp6 [20](S1 Fig). Remarkably, besides the ligand-independent Mg1LysM homodimerization described above, the crystal structure revealed that chitin induces a dimerization of homodimers and, consequently, that a chitin-binding groove is formed by two LysM domains from two independent protomers (Fig 1, Fig 2). In addition to the amino acids that are in direct contact with the chitin trimer, the ligand-induced dimerization is strengthened by several hydrogen bonds that occur between residues from the two protomers involved. One salt bridge between residues K$^{31}$ and D$^{54}$ of the different protomers stabilizes the binding of the single chitin molecule and adds further strength to the dimerization, resulting in a tight binding pocket in which the chitin trimer is strongly bound (Fig 2). Arguably, we would expect an increased chitin-binding affinity of Mg1LysM when compared with a single-acting LysM domain, which can explain in turn why the chitin remained adhered to the Mg1LysM protein during the protein purification procedure.

In order to confirm the chitin-binding activity and determine the Mg1LysM chitin-binding affinity, a polysaccharide affinity precipitation assay and isothermal titration calorimetry (ITC) analysis were used, respectively. Since the crystal structure revealed that a portion of the Mg1LysM binding sites were occupied by chitin in the *P. pastoris*-produced Mg1LysM preparation, we pursued production of Mg1LysM in the bacterium *Escherichia coli* as a heterologous system that is devoid of chitin, in order to obtain chitin-free protein. The precipitation assay confirmed the binding of Mg1LysM to chitin and not to other insoluble polysaccharides [15] (S3 Fig). Subsequent ITC analysis based on chitohexaose (GlcNAc)$_6$ titration revealed that this protein preparation bound chitin with a binding affinity of 4.36 μM (Fig 3). As previously demonstrated [20], *P. pastoris*-produced Ecp6 monomers bind chitin with a stoichiometry of 1:1 (S3 Fig). In contrast, a stoichiometry of 1:2 was observed (n = 0.504) for Mg1LysM based on a single-binding-site model, analogous to the observation that two Mg1LysM protomers originating from two Mg1LysM homodimers bind a single chitin trimer as disclosed in the crystal structure model. Obviously, this ratio also implies a polymerisation reaction in solution upon addition of the ligand chitohexaose. Thus, this finding can be interpreted as an independent confirmation of the ligand-induced Mg1LysM polymerisation as observed in the crystal structure.

## Mg1LysM sequence conservation in a world-wide collection of *Z. tritici* isolates

In order to evaluate Mg1LysM conservation in *Z. tritici*, the occurrence of sequence polymorphisms was evaluated in a collection of 149 isolates from four different populations collected in Switzerland, Australia, Israel and the USA [38]. This analysis revealed that the Mg1LysM protein sequence is highly conserved (S1 Table). Only five non-synonymous mutations were identified in the full length Mg1LysM protein, three of which were previously identified [15]. Interestingly, none of these non-synonymous SNPs localized within the signal peptide, the homodimerization surface, the chitin-binding groove or concerned the residues involved in disulfide or salt bridge formation (Fig 4A), pointing towards the relevance of these sites for the functionality of Mg1LysM. To test the impact of these polymorphisms on chitin binding, we heterologously produced two allelic variants of Mg1LysM in *E. coli*, namely the variants from the *Z. tritici* isolates ST99-CH1E4 and ST99-CH3F4 that, collectively, carry the five non-synonymous mutations (Fig 4C; S1 Table). Whereas both strains share the polymorphisms N3Q and Q34K, strain ST99-CH1E4 additionally carries R24Q, while strain ST99-CH3F4

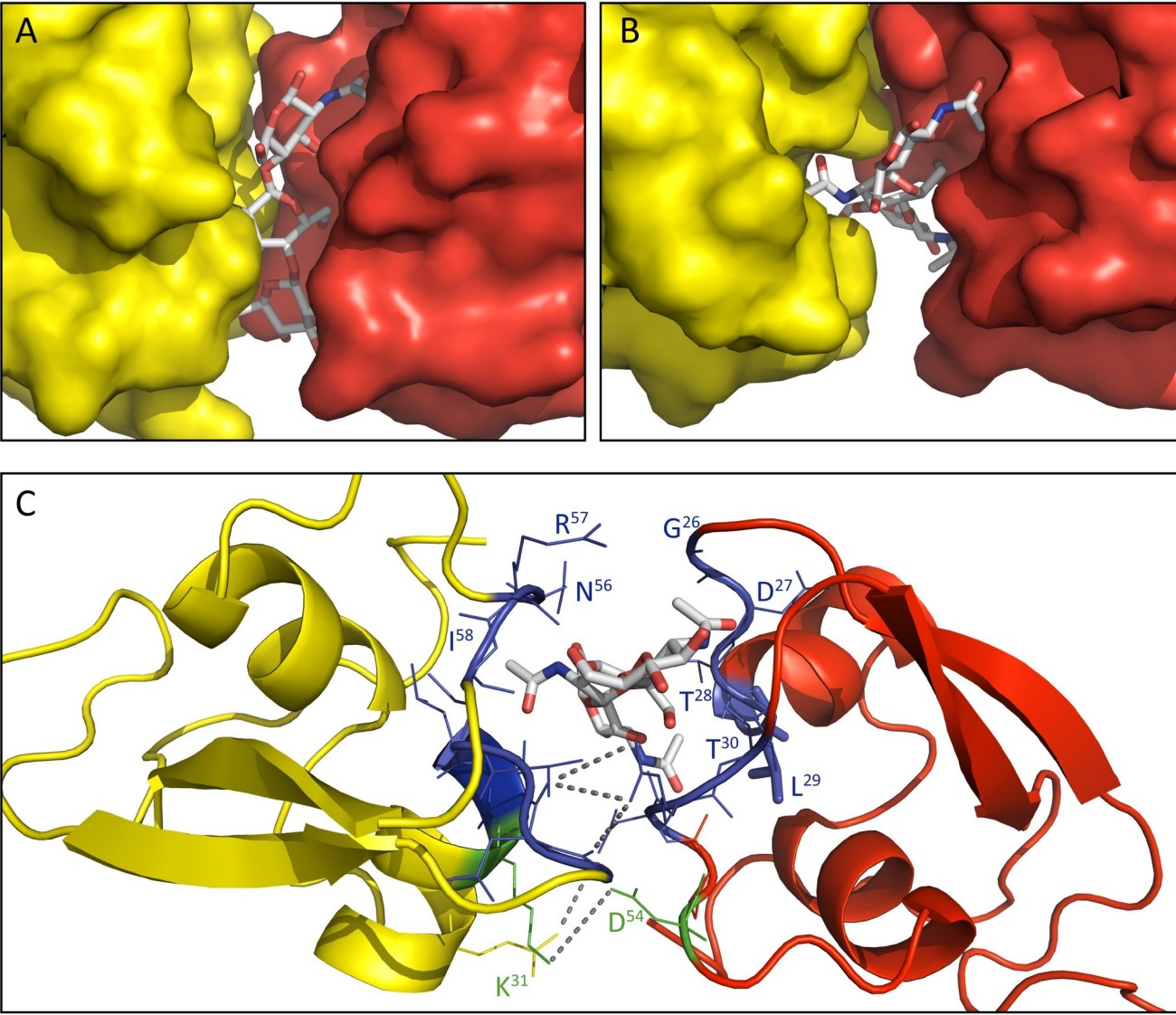

**Fig 2. The chitin binding groove formed by two Mg1LysM protomers.** (A) A chitin trimer (GlcNAc)$_3$, displayed as grey sticks, was identified in a binding pocket formed by two Mg1LysM protomers (indicated in yellow and red, respectively). (B) Representation of the binding pocket from the top. (C) Detail of the chitin binding site. The amino acids involved in direct chitin trimer binding ($^{26}$GDTLT$^{30}$ and $^{56}$NRI$^{58}$) are represented with blue sticks and labelled. In addition, K$^{31}$ and D$^{54}$ (represented in green) of the two different Mg1LysM protomers form a salt bridge that tightly closes the binding pocket. Grey discontinuous lines represent the salt bridge and the hydrogen bonds between the protomers.

additionally carries the polymorphisms R48K and Q20T (Fig 4C; S1 Table). Interestingly, a polysaccharide affinity precipitation assay revealed that, like the wild-type protein Mg1LysM, the two variants Mg1LysM_1E4 and Mg1LysM_3F4 still bind chitin (Fig 4D), suggesting that the allelic variants have retained their biological activity.

## Chitin-induced Mg1LysM polymerization is crucial for protection of hyphae against the hydrolytic activity of plant chitinases

We previously showed that Mg1LysM is able to protect chitin against chitinase hydrolysis [15]. Localization experiments making use of BODIPY-labelled Mg1LysM protein exogenously

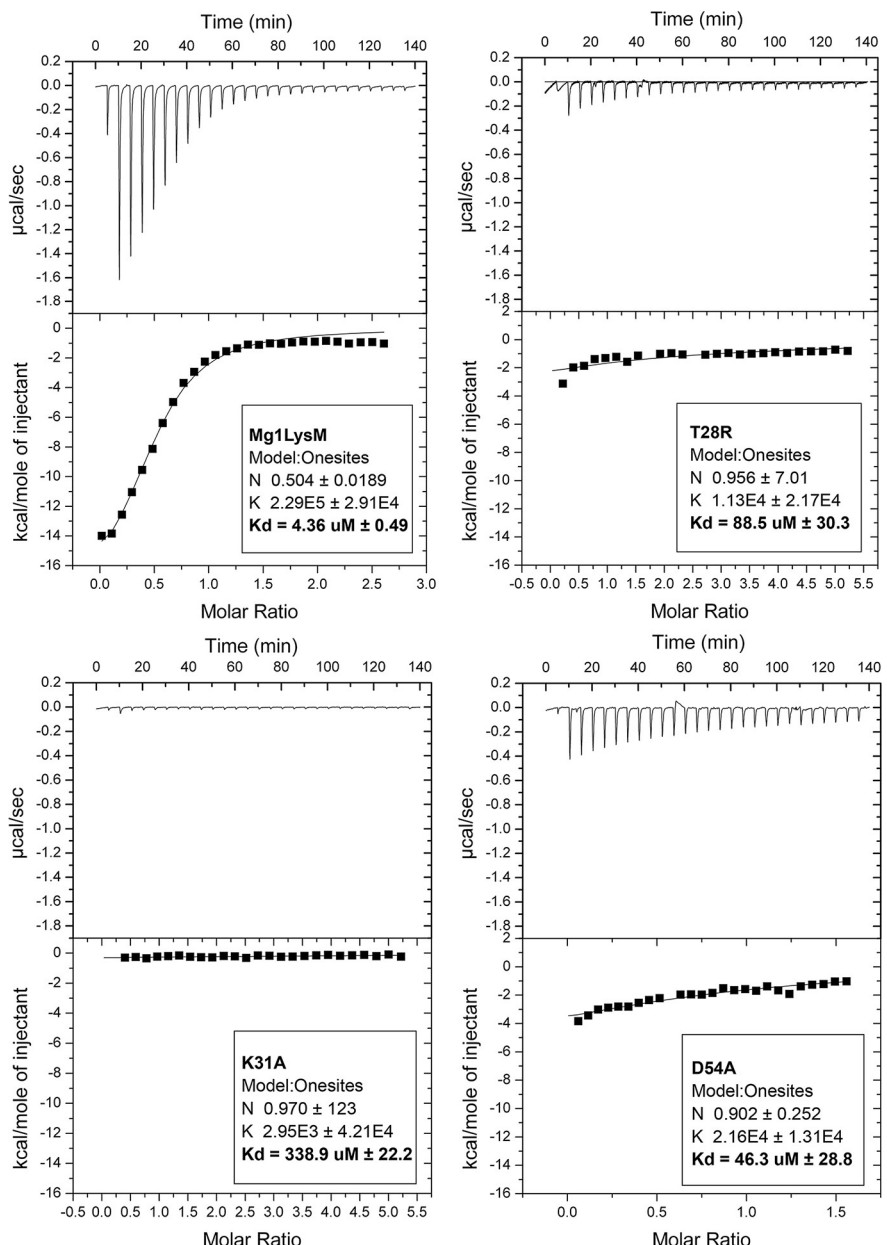

**Fig 3. Two Mg1LysM protomers bind a single chitin hexamer with high affinity.** Isothermal titration calorimetry of $(GlcNAc)_6$ binding by wild-type Mg1LysM produced in *E. coli*, and the mutants $T^{28}R$, $K^{31}A$ and $D^{54}A$. The dissociation constant ($K_d$) and the stoichiometry (N) of the interactions are indicated.

applied to *Trichoderma viride*, a fungal species that exposes cell wall chitin during growth *in vitro*, revealed that Mg1LysM binds to fungal cell walls (Fig 5A). Next, we attempted to evaluate the contribution of the ligand-independent Mg1LysM homodimerization to hyphal protection against chitinases. To this end, we pursued to produce an Mg1LysM mutant that lacked the 12-amino acid tail that is, besides the large protomer-protomer interface, responsible for ligand-independent homodimerization. Unfortunately, production of this mutant in the heterologous host *P. pastoris* was not successful as hardly any protein could be detected. The protein is apparently degraded either due to exposure of the hydrophobic residues (V40 and I68)

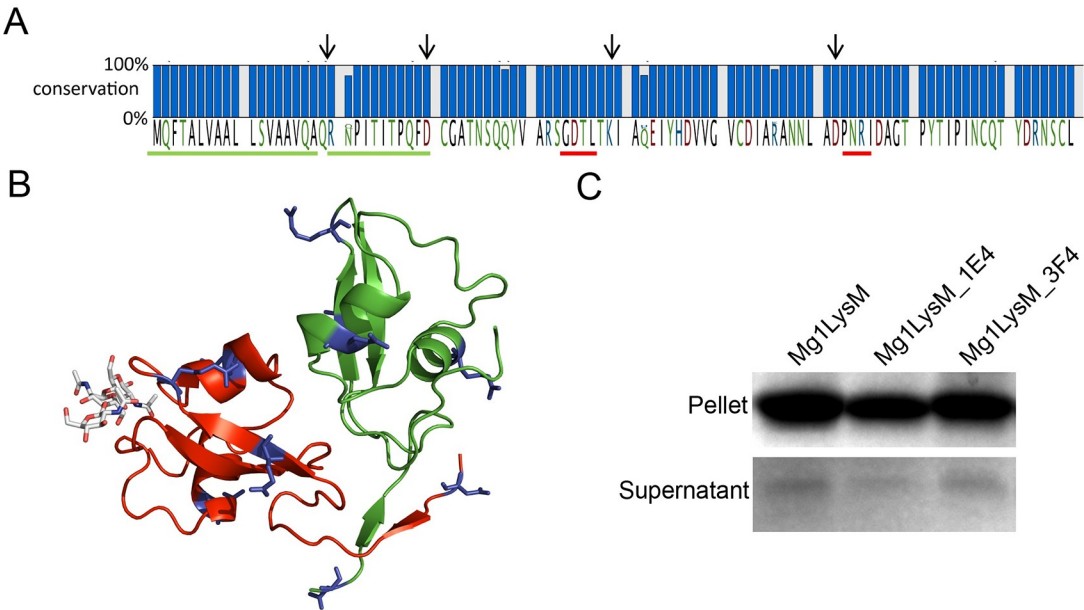

**Fig 4. Mg1LysM sequence polymorphisms in *Z. tritici*.** (A) Five non-synonymous SNPs were identified in 149 *Z. tritici* strains from four different populations. Arrows indicate the position of the residues involved in the formation of salt bridges, while green underlining indicates the signal peptide and red underlining the chitin-binding loops. Red and green underlines indicate the signal peptide and the chitin binding sites, respectively. (B) While the mutations (shown in blue sticks) do not co-localize but occur dispersed over the Mg1LysM protein, none of them is in the chitin-binding site or in the (homo-)dimerization surface. (C) Mg1LysM and the two allelic variants Mg1LysM_1E4 and Mg1LysM-3F4 bind insoluble chitin. All proteins were heterologously produced in *E. coli* and incubated with chitin for 6 hours. After centrifugation, pellets and supernatants were analysed on polyacrylamide gel followed by CBB staining.

located at the centre of the large (1113 Å2) homodimerization interface, or homodimerization is stringently required for proper folding of the protein.

Subsequently, we evaluated the role of chitin-induced Mg1LysM homodimerization in the protection of fungal hyphae against chitinases by generating three mutant proteins. The $T^{28}$ residue that makes direct contact with the chitin substrate in the binding site that was previously shown to be essential for chitin binding in the *C. fulvum* LysM effector Ecp6 was substituted by arginine. In addition, the two residues involved in the formation of the intermolecular salt bridge near the chitin binding site ($K^{31}$ and $D^{54}$) were substituted by alanines, respectively. In order to obtain chitin-free proteins, production in *E. coli* was pursued.

Based on previous findings for Ecp6 [20], we predicted that the $T^{28}R$ mutant was incapable of binding chitin, but the chitin binding capacity of the mutants impaired in the intermolecular salt bridge formation remained to be elucidated [20]. ITC analysis with the mutant $T^{28}R$ revealed a significantly reduced binding affinity of 88.5 μM, which is twenty times weaker when compared with wild-type Mg1LysM protein produced in *E. coli* (4.36 μM; Fig 3). However, also the binding affinity of the mutants $K^{31}A$ and $D^{54}A$ decreased, to 338.9 μM and 46.3 μM, respectively (Fig 3). Furthermore, besides a lower chitin-binding capacity, the stoichiometry calculated for the mutants $K^{31}A$ and $D^{54}A$ changed from 1:2 as observed for the wild-type Mg1LysM protein to 1:1. This finding implies that a single monomer of $K^{31}A$ or $D^{54}A$ binds a single chitohexaose in solution, whereas a single chitohexaose is bound by two wild-type Mg1LysM protomers, supporting the hypothesis that the chitin-induced dimerization is impaired in $K^{31}A$ and $D^{54}A$ by disruption of the intermolecular salt bridge (Fig 3).

Subsequently, we tested the ability of the Mg1LysM mutants to protect fungal hyphae against the hydrolytic activity of plant chitinases. To this end, spores of *Fusarium oxysporum*

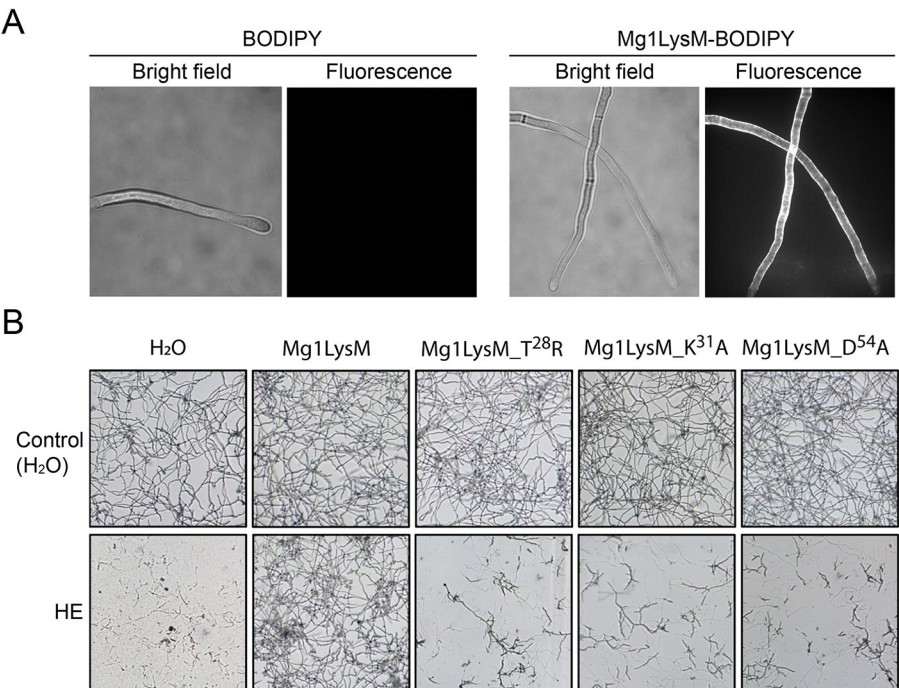

**Fig 5. Mg1LysM mutants are impaired in protection against chitinases.** (A) *Trichoderma viride* incubated with Mg1LysM with the amine-reactive fluorescent dye BODIPY, or with BODIPY only as a control, for 4 hours and bright field and fluorescence microscopy pictures are shown. (B) Microscopic pictures of *Fusarium oxysporum* f. sp. *lycopersici* grown *in vitro* in the absence or presence of wild-type or mutant Mg1LysM, 4 hours after addition of tomato hydrolytic enzymes (HE) that include chitinases, or water as control.

and *Trichoderma viride* were germinated, incubated with a plant extract containing hydrolytic enzymes including chitinases, and supplemented with wild-type or mutant Mg1LysM. As expected, wild-type Mg1LysM protein produced in *E. coli* prevented the hydrolysis of hyphae of *Fusarium oxysporum* f. sp. *lycopersici* (Fig 5B) and *Trichoderma viride* (S4 Fig). Furthermore, mutant T[28]R that is mutated in the substrate-binding loop did not protect *F. oxysporum* and *T. viride* hyphae against these hydrolytic enzymes (Fig 5B and S4 Fig), confirming that chitin-binding of Mg1LysM is required to confer protection of cell walls against hydrolysis by plant enzymes. Considering the even lower chitin-binding activity, it is not surprising that also mutant K[31]A did not protect cell walls against enzymatic hydrolysis. However, also mutant D[54]A no longer protected cell walls, suggesting that a ten-fold reduction of chitin-binding affinity is sufficient to disrupt the protective activity of Mg1LysM. Unfortunately, based on these findings it is impossible to determine the contribution of the dimerization to the protection activity of Mg1LysM.

We previously determined that LysM effector Ecp6 has two sites that bind chitin with 1:1 stoichiometry, one with ultra-high affinity ($k_d$ = 280 pM) and one in the range with which Mg1LysM binds chitin ($k_d$ = 1.70 μM) [13,20,32,33,34,35], but both of which bind chitin with higher affinity than Mg1LysM. Intriguingly, Ecp6 fails to protect hyphae against hydrolysis by chitinases [14,20]. Nevertheless, localization experiments making use of constitutive expression of C-terminally GFP-tagged Ecp6 in *Verticillium dahliae*, and of BODIPY-labelled Ecp6 protein exogenously applied to *Botrytis cinerea*, two fungal species that expose chitin on their cell walls during growth *in vitro*, revealed that Ecp6 can bind to fungal cell walls (S5 Fig) in a similar fashion as *Cladosporium fulvum* effector protein Avr4 that protects fungal cell walls

against hydrolysis by chitin binding through an invertebrate chitin-binding domain [18,19]. These findings suggest that binding of a LysM effector to cell wall chitin with high affinity is not sufficient to mediate protection against hydrolytic enzymes. Moreover, from these observations we infer that chitin-induced dimerization of Mg1LysM may be crucial for hyphal protection against plant enzymatic hydrolysis.

Considering that Mg1LysM homodimers possess two chitin-binding sites on opposite sides of the protein complex (Fig 1), combined with the observed chitin-induced dimerization that may be responsible for the protective activity, we hypothesized that Mg1LysM will form highly oligomeric super-complexes in which ligand-independent Mg1LysM homodimers dimerize on both ends in a chitin-dependent manner (Fig 6A). Moreover, we hypothesized that LysM effectors that do not protect hyphae against chitinase hydrolysis would not display such oligomerisation. To test these hypotheses, we first assessed whether we could alter the particle size distribution of Mg1LysM in solutions by adding chitohexaose. Using dynamic light scattering (DLS) we observed that, upon chitin addition at a molar ratio of 1:2 the radius distribution of Mg1LysM particles shifted from around 10 nm in the absence of chitin to 100 nm in the presence of chitin. Moreover, further increase of the chitin concentration to a 1:5 ratio induced a strong signal appearing at 100 µM, demonstrating clear ligand-induced polymerisation of Mg1LysM protein to large protein complexes. Next, we assessed the effect of chitohexaose on the distribution of Ecp6 particles in solution. Interestingly, although the addition of chitohexaose smoothened the Ecp6 particle size distribution, suggesting the stabilization of Ecp6 molecules, chitin addition did not lead to an increased particle size. Thus, in contrast to Mg1LysM, Ecp6 does not display chitin-induced polymerization.

To confirm our observations with respect to the chitin-induced polymerization of Mg1LysM, we pursued an independent validation. To this end, we reasoned that if Mg1LysM indeed polymerizes in the presence of chitin, we should ultimately be able to precipitate such polymeric complexes during centrifugation into a pellet, whereas such a pellet cannot occur in case polymerization does not take place. Thus, we incubated Mg1LysM overnight with chitohexaose and subsequently centrifuged the sample at 20,000 g in the presence of 0.002% methylene blue to visualize the protein. A clear pellet appeared when Mg1LysM was incubated with chitin, but not in the control treatment where no chitin was added to the Mg1LysM protein, nor in the control treatment with chitohexaose only in the absence of Mg1LysM (Fig 7). Collectively, these data confirm the occurrence of chitin-induced Mg1LysM effector polymers. In contrast, similar treatments of Ecp6 with chitin did not result in a pellet after centrifugation, confirming that Ecp6 does not undergo chitin-induced polymerization (Fig 7).

Recently, it was demonstrated that the arbuscular mycorrhizal (AM) fungus *Rhizophagus irregularis* secretes the LysM effector RiSLM to facilitate its mutualistic symbiosis [39]. Interestingly, like Mg1LysM, RiSLM was shown to be composed of a single LysM only, and furthermore to protect hyphae against hydrolysis by chitinases. Thus, we used RiSLM to test whether chitin-induced polymerisation is restricted to Mg1LysM only, or similarly occurs for RiSLM as well. Interestingly, like with Mg1LysM, a chitin-induced particle size shift was observed in the DLS assay with RiSLM (Fig 6B). Moreover, upon overnight incubation of RiSLM with chitohexaose a clear pellet could be obtained after centrifugation, demonstrating that polymerization occurred (Fig 7). Collectively, these data suggests that chitin-induced polymerization is a common phenomenon that occurs not only with Mg1LysM, but also with other LysM effectors that protect fungal hyphae.

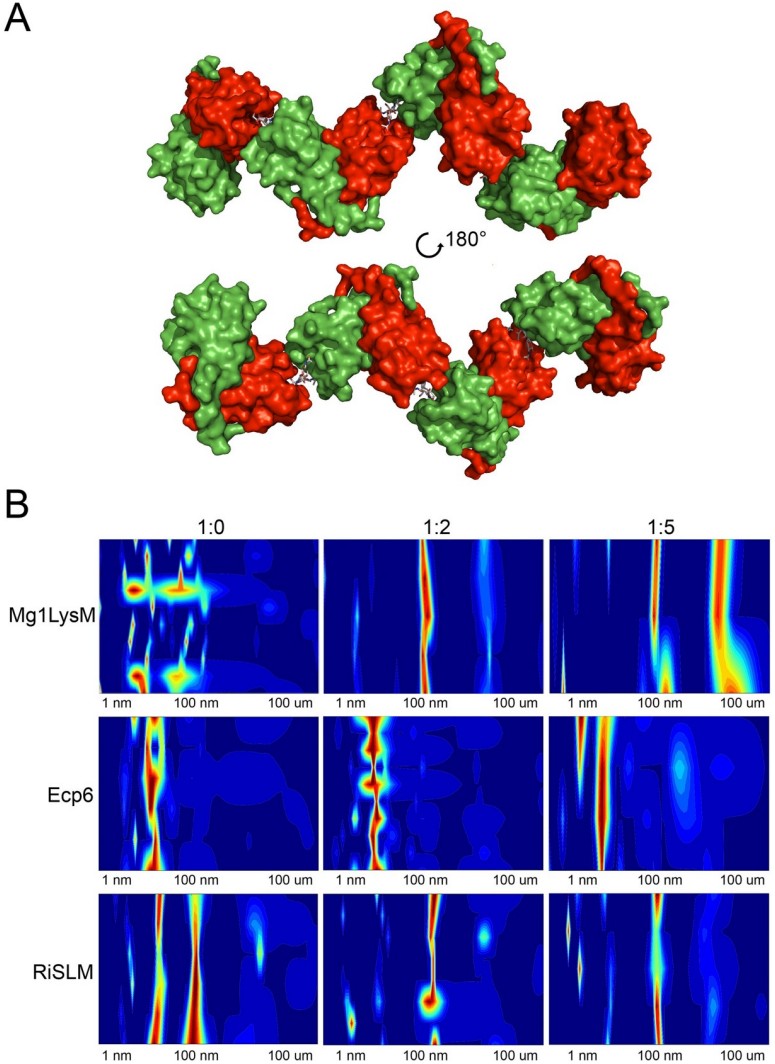

**Fig 6. Chitin-induced polymerization of Mg1LysM homodimers.** (A) Model inferred from the crystal structure of Mg1LysM in which a continuous structure of Mg1LysM homodimers and chitin is formed. Alternating chitin molecules (in grey sticks) and Mg1LysM homodimers (in red and green), each of them with two chitin-binding sites, are shown. (B) Dynamic light scattering (DLS) heat maps of Mg1LysM, *C. fulvum* Ecp6 and RiSLM treated with chitohexaose in molar ratios of 1:0, 1:2 and 1:5 (protein: chitohexaose), respectively. The particle size distribution is indicated as a color scale ranging from blue (lowest amount) to red (highest amount) for a particle size range of 1 nm to 100 um.

## Discussion

Studies on many plant pathogenic fungi have shown that the perception of microbial cell wall-derived glycans by plant hosts plays a central role in microbe–host interactions [10]. Among these glycans, fungal cell wall chitin has emerged as one of the most potent fungal elicitors of host immune responses [3,10]. The widespread glycan perception capacity in plants has spurred the evolution of various fungal strategies to evade glycan perception [3,10]. Many fungal pathogens secrete LysM effectors to perturb the induction of chitin-triggered immunity. Structural analysis of the *C. fulvum* LysM effector Ecp6 has revealed that this activity could be attributed to the presence of an ultra-high chitin binding affinity site in the LysM effector protein that is established by intramolecular LysM domain dimerization [20]. However, some

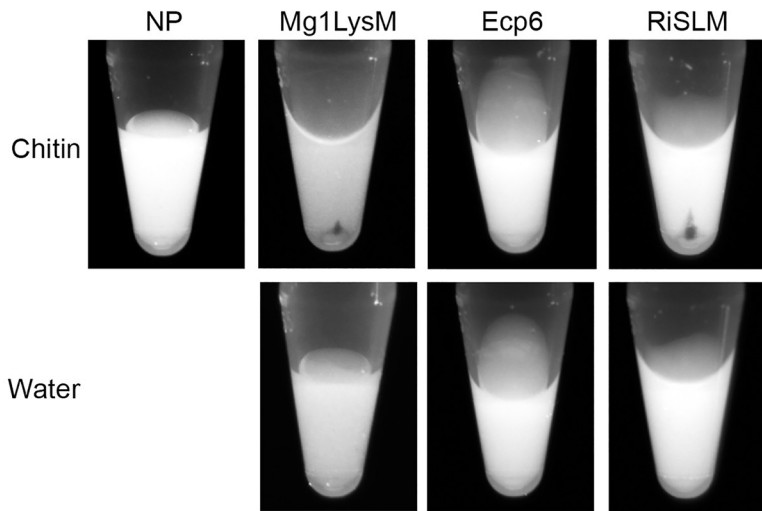

**Fig 7. Chitin-induced polymerization of Mg1LysM and RiSLM, but not of Ecp6.** Effector proteins were incubated with chitohexaose (chitin) or water as control. After overnight incubation, methylene blue was added and protein solutions were centrifuged, resulting in protein pellets as a consequence of polymerization for Mg1LysM and RiSLM, but not for Ecp6. NP is the 'no protein' sample with chitin in the absence of LysM effector protein.

LysM effectors rather, or additionally, are able to prevent the hydrolysis of fungal cell wall chitin by plant chitinases. Moreover, functional characterization of Mg1LysM, a LysM effector that is merely composed of a single LysM domain and a few additional amino acids, suggested that the ability to protect cell walls is conferred simply by the chitin-binding ability of the LysM domain [15]. Yet, the observation that various other LysM effectors, including *C. fulvum* Ecp6, *M. oryzae* Slp1 and *C. higginsianum* ELP1 and ELP2, are not able to protect hyphae challenged this hypothesis [14,29,30]. Thus, the mechanism by which some LysM effectors are able to protect fungal cell walls remained to be characterized. The crystal structure model that was generated in this study has revealed that Mg1LysM is able to undergo chitin-mediated dimerization such that a chitin molecule is deeply buried in the protein dimer. Nevertheless, the structure of the dimer allows a chitin chain to protrude into the solvent on either side of the binding groove, such that it is conceivable that the dimer can be formed on long-chain chitin polymers of any length, including polymeric cell wall chitin. In addition to several noncovalent bounds between the two Mg1LysM protomers and between the protein and the chitin, a salt bridge between the two Mg1LysM protomers strengthens the chitin-binding affinity and thus supports the chitin-induced dimerization by stabilizing the chitin binding groove. Arguably, it is this particular trait that confers the ability to protect hyphae against plant chitinases, as disruption of the ion bond in the $K^{31}A$ mutant abolished hyphal protection and chitin binding by itself is not sufficient to confer cell wall protection. The crystal structure further revealed that Mg1LysM undergoes ligand-independent homodimerization whereby a large dimerization interface of two Mg1LysM monomers is stabilized by several noncovalent bounds and further strengthened by two salt bridges between the interlaced N-terminal regions of the protein. Despite various efforts, we have not been able to produce monomeric Mg1LysM protein, suggesting that ligand-independent homodimerization is required for proper folding of the protein. Consequently, Mg1LysM homodimers are released and possess two chitin-binding sites on opposite sides of the protein (Fig 1). Combined with the observed chitin-induced dimerization, we postulate that this provides Mg1LysM with the ability to form highly oligomeric super-complexes in the fungal cell wall, in which ligand-independent Mg1LysM homodimers

dimerize on both ends in a chitin-dependent manner, leading to the formation of a contiguous structure throughout the cell wall (Fig 6A). Indeed, a chitin-inducible shift in particle size could be demonstrated for Mg1LysM in DLS experiments, confirming the occurrence of polymerisation, whereas such shift was not observed for Ecp6. Moreover, overnight incubation of Mg1LysM with chitohexaose led to the formation of polymers that could be pelleted during centrifugation, demonstrating that the polymers grew to relatively large particles. Also this phenomenon was not observed for Ecp6. Intriguingly, the recently characterized LysM effector RiSLM from the mutualistic fungus *Rhizophagus irregularis* [39] that, like Mg1LysM, is composed of a single LysM only and protects hyphae against chitinases also showed the ability to polymerize in a chitin-dependent manner, suggesting that polymerization of into oligomeric super-complexes in the fungal cell wall is a common phenomenon among LysM effectors that protect fungal hyphae. Possibly, it is such contiguous structure that provides steric hindrance that renders fungal cell wall chitin inaccessible to chitinase enzymes. Thus, both ligand-independent homodimerization as well as ligand-induced dimerization of Mg1LysM appear to be required for its cell wall protective function. Accordingly, residues shaping these regions are fully conserved in all Mg1LysM isoforms that have been identified to date.

## Methods

### Protein production and purification

Mg1LysM was previously produced in *Pichia pastoris* strain GS115 as described [15]. Purification was performed by gel filtration chromatography (Superdex 75: GE Healthcare, Chicago, IL, US) in 20 mM HEPES, pH 7.0, and 50 mM NaCl. *P. pastoris* produced protein (6–10 mg/mL) was used for protein crystallization. *Escherichia coli* protein production was performed using pET-SUMO (Thermo Fisher, Waltham, MA, USA). Mg1LysM was cloned into pGEM-T (Promega, Madison, WI, US) using specific primers (S2 Table). Mutants were obtained by PCR using overlapping primers with the corresponding mismatch (S2 Table) followed by digestion of the template by *Dpn*I. For the cloning of ST99_CH3F4 and ST99_CH1E4 versions of Mg1LysM, the sequence was commercially synthesized (Eurofins Genomics, Ebersberg, Germany). After digestion of the resulting vector with *Sac*I and *Hind*III (Promega, Madison, WI, US), Mg1LysM was cloned in the final vector pET-SUMO. The expression system *E. coli* ORIGAMI (DE3, Merck, Darmstadt, Germany) cells was used to express the protein following the manufacturer´s instructions. Transformants were selected and grown in Luria broth (LB) medium until an optical density of 0.8 at 600 nm was reached. Protein production was induced with the addition of 0.05 mM Isopropyl ß-D-1-thiogalactopyranoside (IPTG) at 28˚C. Cells were harvested by centrifugation ~20 h after induction, the cell pellets were dissolved and lysed using lysozyme from chicken egg (Sigma-Aldrich, St. Louis, MO, US) and the 6xHis-SUMO tagged proteins were purified from the soluble protein fraction after centrifugation using an $Ni^{2+}$-NTA Superflow column (Qiagen, Venlo, Netherlands). Next, purified proteins were incubated with the SUMO protease ULP1 from *Saccharomyces cerevisiae* (Thermo Fisher, Waltham, MA, USA), dialysed over night against 200 mM NaCl at 4˚C, and again passed through a $Ni^{2+}$-NTA Superflow column. Native proteins were finally dialysed against 50 mM $NaH_2PO_4$, 300 mM NaCl at pH 8.0, concentrated to 0.6 mg/mL over Amicon ultracentrifugal filter units (Sigma-Aldrich, St. Louis, MO, USA) and used for subsequent assays. RiSLM was produced as described previously [39].

### Crystallization conditions and structure determination

First crystal hits with 1,4-dioxane as the reservoir solution were obtained overnight in a small initial vapor-diffusion crystallization screening campaign using the Phoenix robot (Art

Robbins Instrument LLC, Sunnyvale, CA, USA) with 96-well Intelli Plates (Dunn Labortechnik GmbH, Asbach, Germany) and several different commercial screens (Hampton Research, Aliso Viejo, CA, USA; Molecular Dimensions, Newmarket, Suffolk, UK) [40]. Conditions were further optimized and useful crystals were finally obtained by micro-seeding techniques using 0.1 M sodium citrate pH 5.6, 5%-20% PEG4000 and 5% isopropanol as the reservoir solution [31]. 0.2 M sodium acetate pH 4.6 with 20% ethylene glycol was used as the crystal cryo-buffer. Several crystals were soaked with either I3C (Jena Bioscience GmbH, Jena, Germany), 2 mM in cryo-buffer, quick soak, or $Ta_6Br_{14}$ (Jena Bioscience GmbH, Jena, Germany), 1 mM in cryo-buffer, 1 hr soak, brief wash and prolonged back soak. X-ray diffraction data were collected on BL14.1 at the BESSY II electron storage ring operated by the Helmholz-Zentrum Berlin [41]. Using the Phenix AutoSol wizard [42], initial phases were obtained from the $Ta_6Br_{12}{}^{2-}$ derivatized crystals by single-wavelength anomalous dispersion techniques (SAD) that were improved by phase information from the I3C derivatized crystals by single isomorphous replacement with anomalous scattering (SIRAS).

The structure was refined using *REFMAC*5 [43] and phenix [42] and manually built using *Coot* [44]. All figures showing structural representations were prepared with the program *PyMOL* (The PyMOL Molecular Graphics System, Version 2.0 Schrödinger, LLC, DeLano Scientific, Palo Alto, CA, USA]. The quality of the final model was validated with *MolProbity* [45]. Refinement and phasing statistics are summarized in Table 1.

## Chitinase-protection assay

*In-vitro* chitinase protection assays were performed as described previously [21]. Essentially, $\sim 10^3$ conidiospores of *Fusarium oxysporum* f. sp. *lycopersici* or *Trichoderma viride* were incubated overnight at room temperature in 40 μL of half-strength potato dextrose broth (PDB; Becton Dickinson, Franklin Lakes, NJ, USA) in a 96-well microtiter plate. Subsequently, wild-type or mutant Mg1LysM protein was added at a final concentration of 20 μM. After a 2 h incubation period, 10 μL of tomato extract containing hydrolytic enzymes was added [21]. Fungal growth was assessed microscopically after 4 h of incubation at room temperature.

## Polysaccharide precipitation assay

The polysaccharide precipitation assay was performed as described [15]. 800 μL of Mg1LysM (30 μg/mL) was incubated with 50 μL of chitin beads (NEB, Massachusetts, USA), 10 mg shrimp chitin, chitosan, cellulose or xylan (all from Sigma-ALRICH, Missouri, USA) for 6 h at 4°C. The insoluble fraction was pelleted by centrifugation (13,000 g, 5 min), resuspended in 100 μL of water and boiled at 95°C for 10 minutes. The supernatant was concentrated using Microcon Ultracel YM-10 tubes (Merck, Darmstadt, Germany) to 80 μL, boiled at 95°C for 10 minutes with 30 μL of protein loading buffer (4×). The presence of proteins in pellet and supernatant was examined on a Mini-PROTEAN TGX Stain-Free Gel (Bio-Rad, California, USA) followed by Coomassie Brilliant Blue staining.

## Isothermal titration calorimetry

Isothermal titration calorimetry (ITC) experiments were performed at 20°C following standard procedures using a Microcal VP-ITC calorimeter (GE Healthcare, Chicago, IL, US). The *E. coli*-produced wild-type Mg1LysM (20 μM) and the mutants $T^{28}R$ (15 μM), $K^{31}A$ (30 μM) and $D^{54}A$ (30 μM) were titrated with a single injection of 2 μL, followed by 26 injections of 10 μL of $(GlcNAc)_6$ (Isosep AB, Tullinge, Sweden) at 200 μM. Ecp6 (15 μM) was titrated with $(GlcNAc)_6$ at 400 μM. Before the experiment, all proteins were dialyzed against 20 mM of sodium chloride, pH 7.0. Chitohexaose (Megazyme, Wicklow, Ireland) was freshly dissolved

in the dialysis buffer. Data were analyzed using Origin (OriginLab, Northampton, MA, USA) and fitted to a one-binding-site model. Before and after the experiment protein samples were analysed on SDS-PAGE gel and stained with Coomassie Brilliant Blue (S3C Fig).

## Dynamic light scattering (DLS) measurements

Mg1LysM, RiSLM and Ecp6 were dialyzed overnight against water, and subsequently incubated with 0.01% Triton X-100 for 4 hours to improve protein solubility. Next, chitohexaose (Megazyme, Wicklow, Ireland) was added in a molar ratio of 1:0, 1:2 and 1:5 (protein:chitin) and incubated overnight. Particle size distribution was measured by a SpectroSize 300 (Xtal Concepts, Hamburg, Germany).

## Polymerization assay

Mg1LysM and Ecp6 were adjusted to a concentration of 400 μM and 100 μL of each protein was incubated with 100 μL of 4 mM chitohexaose (Megazyme, Wicklow, Ireland), or 100 μL water as control, at room temperature overnight. Similarly, 100 μL of RiSLM (600 μM) was incubated with 75 μL of 4 mM chitohexaose (Megazyme, Wicklow, Ireland) or water as control in a total volume of 200 μL. The next day, 2 μL of 0.2% methylene blue (Sigma-Aldrich, Missouri, USA) was added and incubated for 10 min after which protein solutions were centrifuged at 20,000 g for 15 min. Photos were taken with a ChemiDoc MP system (Bio-Rad, California, USA) with custom setting for red fluorescent protein (RFP).

## Localisation of Mg1LysM and Ecp6

Labelling of effector proteins with BODIPY TMR-X amine reactive probe (Invitrogen, Carlsbad, CA, USA) was performed as described previously [18,19]. For localisation assay of Mg1LysM, conidiospores of *Trichoderma viride* were harvested from five-day-old potato dextrose agar (PDA) plates and adjusted to $10^6$ conidiospores/mL with half-strength PDB. The conidiospore solution was pipetted into a 96-well microtiter plate in aliquots of 50 μL and the plate was incubated overnight at room temperature for germination. The next day, BODIPY-labeled Mg1LysM was applied at a final concentration of 8 μM and incubated for 4 hrs at room temperature in the dark. Microscopic analysis was performed using a Nikon Eclipse Ti microscope using a 100× Plan apo oil immersion objective (NA 1.4) and a 561 nm laserline. Pictures were processed and analysed with ImageJ (http://rsbweb.nih.gov/ij/).

For localisation study of Ecp6, conidiospores of a *V. dahliae* transformant were grown in a few micro liters of PDB on a glass slide with coverslip. To prevent the samples from drying out, the slides were mounted on top of moistened tissue in an empty pipette box with water on the bottom. After approximately 6 hr of growth at room temperature, the slides were used for localization studies. Conidiospores of *Botrytis cinerea* were harvested and germinated overnight in PDB at room temperature. BODIPY-labeled proteins were applied at a concentration of 4 μM and incubated for 2–3 hrs at room temperature in the dark. The localisation studies were performed using a Nikon eclipse 90i UV microscope and NIS-Elements AR 2.3 software (Nikon Instruments Inc., Melville, USA).

## Assembly and alignment of *Mg1LysM* sequences

Illumina whole-genome sequencing data from a global collection of *Z. tritici* isolates was used to extract *Mg1LysM* sequences [46]. We used the SPAdes assembler version 3.6.2 [47] to generate *de-novo* genome assemblies. The SPAdes pipeline includes the BayesHammer read error correction module to build contigs in a stepwise procedure based on increasing k-mer lengths.

We defined the k-mer range as 21, 35, 49, 63 and 77. We used the "—careful" option to reduce mismatches and indel errors in the assembly. Polished assemblies were then used to retrieve the contigs containing *Mg1LysM* orthologs based on blastn [48]. High-confidence sequence matches were extracted with samtools [49] from each draft assembly and aligned using MAFFT version 7.305b [50] using the—auto option and 1,000 iterative refinement cycles. Alignments were processed using JalView [51] and CLC Genomic Workbench 9 (Qiagen, Venlo, Netherlands).

## Supporting information

**S1 Fig. Protein alignment of Ecp6 and Mg1LysM.** (A) Protein sequence alignment of Ecp6 and Mg1LysM. The two chitin binding loops of Mg1LysM are indicated with a blue line and the signal peptide with a green line. Red and blue asterisks indicate the position of the residues involved in the formation of salt bridges in the binding groove and in the dimerization surface, respectively. (B) Structural alignment of the LysM1 domain from Ecp6 (in blue) and the LysM domain from Mg1LysM (in red). The chitin trimer is shown in grey sticks. The chitin binding loops are shown in dark blue and green for LysM1 and for Mg1LysM, respectively. (C) Chitin binding pocket formed by LysM1 and LysM3 of Ecp6. In orange ribbons, a single molecule of Ecp6 is shown. The residues involved in chitin binding are shown as blue sticks. Hydrogen bonds between the two LysM domains are shown in grey. (D) Protein sequence alignment of the LysM domains of Ecp6 and Mg1LysM.
(TIF)

**S2 Fig. $2|F_0|-|F_c|$ map.** $2|F_0|-|F_c|$ electron density map around the chitin trimer (carbon atoms coloured yellow) is contoured at 1 sigma above the mean. Amino acids of the chitin binding motif ($^{26}$GDTLT$^{30}$ and $^{56}$NRI$^{58}$) are represented as sticks (carbon atoms coloured light-grey).
(TIF)

**S3 Fig.** (A) Isothermal titration calorimetry of $(GlcNAc)_6$ binding by Ecp6 produced in *P. pastori*. (B) Mg1LysM protein binds to insoluble chitin, but not to other carbohydrates. The purified Mg1LysM protein produced in *E. coli* was incubated with chitin beads, the insoluble carbohydrates shrimp chitin, chitosan, cellulose and xylan and centrifuged. Both the pellet and the supernatant were analyzed on protein gels. (C) Coomassie Brilliant Blue stained gel of Mg1LysM and mutant proteins before and after ITC assay. 1 and 2 indicate two independent ITC mesurements.
(TIF)

**S4 Fig. Mg1LysM mutants are impaired in protection of *Trichoderma viride* against chitinases.** Microscopic pictures of *Trichoderma viride* grown *in vitro* in the absence or presence of wild-type or mutant Mg1LysM, 4 hours after addition of tomato hydrolytic enzymes (HE) that include chitinases, or water as control.
(TIF)

**S5 Fig. *C. fulvum* LysM effector Ecp6 protein localizes to fungal cell walls.** (A) Brightfield image (left), fluorescence image (middle) and the overlay image (right) of a hypha from an *Ecp6-GFP* transformant of *Verticillium dahliae*. The chitin-binding *C. fulvum* LysM effector Ecp6 (B) and chitin-binding effector Avr4 (C) that carries an invertebrate chitin-binding domain were labeled with the amine-reactive fluorescent dye BODIPY and incubated with *Botrytis cinerea* spores for 2–3 hours and observed with fluorescence microscopy.
(TIF)

**S1 Table. Mg1LysM protein alignment.**
(XLSX)

**S2 Table. Primers used in this study.**
(DOCX)

## Acknowledgments

The authors acknowledge support from the partners of the European Research Area Network for Coordinating Action in Plant Sciences (ERA-CAPS) consortium "SIPIS".

## Author Contributions

**Conceptualization:** Jeroen R. Mesters, Bart P. H. J. Thomma.

**Formal analysis:** Andrea Sánchez-Vallet, Hui Tian, Luis Rodriguez-Moreno, Dirk-Jan Valkenburg, Raspudin Saleem-Batcha, Stephan Wawra, Anja Kombrink, Leonie Verhage, Ronnie de Jonge, H. Peter van Esse, Alga Zuccaro, Daniel Croll.

**Funding acquisition:** Alga Zuccaro, Bart P. H. J. Thomma.

**Investigation:** Andrea Sánchez-Vallet, Hui Tian, Luis Rodriguez-Moreno, Dirk-Jan Valkenburg, Raspudin Saleem-Batcha, Stephan Wawra, Anja Kombrink, Leonie Verhage, Ronnie de Jonge, H. Peter van Esse, Alga Zuccaro, Daniel Croll.

**Methodology:** Andrea Sánchez-Vallet, Hui Tian, Luis Rodriguez-Moreno, Ronnie de Jonge, H. Peter van Esse.

**Supervision:** Jeroen R. Mesters, Bart P. H. J. Thomma.

**Writing – original draft:** Andrea Sánchez-Vallet, Hui Tian, Bart P. H. J. Thomma.

**Writing – review & editing:** Andrea Sánchez-Vallet, Hui Tian, Luis Rodriguez-Moreno, Jeroen R. Mesters, Bart P. H. J. Thomma.

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
