## [Decision Letter · Decision Letter 0]

2 Dec 2019

Dear Dr. Thomma,

Thank you very much for submitting your manuscript "A secreted LysM effector protects fungal hyphae through chitin-dependent homodimer polymerization" (PPATHOGENS-D-19-01885) for review by PLOS Pathogens. Your manuscript was fully evaluated at the editorial level and by independent peer reviewers. The reviewers appreciated the attention to an important problem, but raised some substantial concerns about the manuscript as it currently stands. The authors should provide more biochemical evidences to support the conclusion before we would be willing to consider a revised version of your study. We cannot, of course, promise publication at that time.

We therefore ask you to modify the manuscript according to the review recommendations before we can consider your manuscript for acceptance. Your revisions should address the specific points made by each reviewer.

(1) A letter containing a detailed list of your responses to the review comments and a description of the changes you have made in the manuscript. Please note while forming your response, if your article is accepted, you may have the opportunity to make the peer review history publicly available. The record will include editor decision letters (with reviews) and your responses to reviewer comments. If eligible, we will contact you to opt in or out.

(2) Two versions of the manuscript: one with either highlights or tracked changes denoting where the text has been changed; the other a clean version (uploaded as the manuscript file).

Additionally, to enhance the reproducibility of your results, PLOS recommends that you deposit your laboratory protocols in protocols.io, where a protocol can be assigned its own identifier (DOI) such that it can be cited independently in the future. For instructions see http://journals.plos.org/plospathogens/s/submission-guidelines#loc-materials-and-methods

We hope to receive your revised manuscript within 60 days. If you anticipate any delay in its return, we ask that you let us know the expected resubmission date by replying to this email. Revised manuscripts received beyond 60 days may require evaluation and peer review similar to that applied to newly submitted manuscripts.

There is additional documentation related to this decision letter. To access the file(s), please click the link below. You may also login to the system and click the 'View Attachments' link in the Action column.

[LINK]

Sincerely,

Hui-Shan Guo

Associate Editor

PLOS Pathogens

Wenbo Ma

Section Editor

PLOS Pathogens

Kasturi Haldar

Editor-in-Chief

PLOS Pathogens

orcid.org/0000-0001-5065-158X

Grant McFadden

Editor-in-Chief

PLOS Pathogens

orcid.org/0000-0002-2556-3526

Reviewer's Responses to Questions

**Part I - Summary**

Reviewer #1: This study by Sánchez-Vallet et al. addressed on the crystal structure and biological function of the LysM effector Mg1LysM during the microbe-host interaction. The authors determined the crystal structure of the Mg1LysM and proposed a model for how Mg1LysM protects hyphae from host chitinase. Mg1LysM can form the chitin-dependent dimer and chitin-independent homodimer. The Mg1LysM protects hyphae from chitinase by formation of a supramolecular structure by chitin-induced oligomerization of the Mg1LysM homodimers. In general, this work gives a novel insight into the study of fungal LysM effectors. However, the conclusions are mostly based on the crystal structure data, more experiments are required to support the author’s conclusions.

Reviewer #2: (No Response)

Reviewer #3: In this manuscript the authors presented a crystal structure of the single LysM domain-containing effector Mg1LysM of the wheat pathogen Zymoseptoria tritici, showing that chitin-induced dimerization of two Mg1LysM protomers through high affinity binding is required for hyphal protection against chitinases. The structure shows Mg1LysM also forms ligand independent homodimers, suggesting a supramolecular structure can be formed in which chitin-induced oligomerization of Mg1LysM ligand-independent homodimers form a contiguous Mg1LysM higher ordered structure that is anchored to the chitin in the fungal cell wall to prevent hydrolysis by host chitinases. There are some critical notes on the characterization of the dimers and their function in fungal cell wall. Therefore, I recommend this manuscript be published in PLOS Pathogens Journal after major revision as shown below.

**Part II – Major Issues: Key Experiments Required for Acceptance**

Reviewer #1: 1. In figure 3, when examining Mg1LysM chitin-binding affinity by ITC assay, ECP6 should be included as a positive

control.

A second assay (e. g. in vitro pull-down) is also required to further confirm the Mg1LysM chitin-binding affinity.

2. In figure 6, it would be more convincible if the authors run the gel filtration chromatography to prove the chitin-

induced Mg1LysM oligomerization and determine the molecular weight of the super-complex.

3. In Fig S4, the authors checked the subcellular location of ECP6 and Avr4, I think it is more important to examine

the subcellular location of LysM1.

Reviewer #2: As commented in the review letter, biochemical evidence should be provided to further support the existence of polymerzation of Mg1LysM in solution. To demonstrate the biological significance of Mg1LysM polymerzation of, the authors need to identify mutations that disrupt or impair the polymerzation lower the activity of protecting hyphae against the hydrolytic activity of plant chitinases.

Reviewer #3: 1. The claim of authors about D54A mutant no longer protected cell walls, suggesting that a ten-fold reduction of chitin-binding affinity is sufficient to disrupt the protective activity of Mg1LysM is skeptical. For example, why the disruption of a same pair of salt bridge (D54-K31) causes a much higher difference (338.9 μM vs 46.3 μM) in their chitin-binding affinities.

2. It was hypothesized that LysM effectors that do not protect hyphae against chitinase hydrolysis would not display oligomerisation. The hypothesis was tested by DLS using Ecp6 in comparison to Mg1LysM. Analysis of additional LysM effectors is required to strengthen the hypothesis furthermore.

3. This manuscript claimed that the disruption of ligand-independent homo-dimer would result in loss of function for Mg1LysM. However, more direct evidence is needed to show such functional relevance.

**Part III – Minor Issues: Editorial and Data Presentation Modifications**

Reviewer #1: 1. To confirm the functional conservation of Mg1LysM1, the authors should check the chitin-binding affinity of

several other Mg1LysM homologs from other isolates.

2. In figure 1A, the recombinant Mg1LysM protein used for determination of the crystal structure was

heterologously produced in the yeast Pichia pastoris. The authors should present a figure to show the protein

expression of Mg1LysM.

3. In line 160, there’s no figure 1B and 1C. please check it.

4. In figure 3, the recombinant protein used for the ITC assays should be shown by CBB staining.

A second assay (e. g. in vitro pull-down) are required for confirm the Mg1LysM-chitin interaction. Ecp6 should be

include as a positive control.

Reviewer #2: (No Response)

Reviewer #3: (No Response)

PLOS authors have the option to publish the peer review history of their article (what does this mean?). If published, this will include your full peer review and any attached files.

Reviewer #1: No

Reviewer #2: No

Reviewer #3: No

---

## [Decision Letter · Decision Letter 1]

17 May 2020

Dear Dr. Thomma,

Thank you very much for submitting your manuscript "A secreted LysM effector protects fungal hyphae through chitin-dependent homodimer polymerization" for consideration at PLOS Pathogens. As with all papers reviewed by the journal, your manuscript was reviewed by members of the editorial board and by several independent reviewers. 

The revision was evaluated again by the same reviewers who evaluated the original submission. Although two reviewers were satisfied with the revision, Reviewer 2 has remaining concerns. In light of the reviews (below this email), we would like to invite the resubmission of a revised version that takes into account the reviewer' comments.

We cannot make any decision about publication until we have seen the revised manuscript and your response to the reviewers' comments. Your revised manuscript is also likely to be sent to reviewers for further evaluation.

Sincerely,

Hui-Shan Guo

Associate Editor

PLOS Pathogens

Wenbo Ma

Section Editor

PLOS Pathogens

Kasturi Haldar

Editor-in-Chief

PLOS Pathogens

orcid.org/0000-0001-5065-158X

Michael Malim

Editor-in-Chief

PLOS Pathogens

orcid.org/0000-0002-7699-2064

Reviewer's Responses to Questions

**Part I - Summary**

Reviewer #1: The present manuscript by Sánchez-Vallet et al. determined the crystal structure of Mg1LysM1 and proposed a model for how Mg1LysM protects hyphae from host chitinase, which gives a novel insight into the study of fungal LysM effectors. The authors have now provided sufficient biological and biochemical experiments to support the working model and replied most of the reviewer's concerns. Thus, I recommend this work published on PLos Pathogens.

Reviewer #2: (No Response)

Reviewer #3: The authors have addressed most of my previous concerns and comments. To further support their hypothesis, the authors tested another effector RiSLM from fungus Rhizophagus, and showed similar polymerisation in a chitin-dependent manner. The author wanted to investigate how the disruption of ligand-independent homo-dimer would lead to loss of function in Mg1LysM. Unfortunately, they failed to produce mutant proteins, possibly due to the protein instability. However, they explained the more details about the issue in the result.

Overall, this manuscript is much improved, and should be be acceptable for publication.

**Part II – Major Issues: Key Experiments Required for Acceptance**

Reviewer #1: (No Response)

Reviewer #2: In the revision, the authors performed a polymerization assay in order to support chitin-induced oligomerzation of Mg1LysM. Unfortunately, the data from the assay are not sufficient to support the conclusion. They must present direct biochemical evidence for this using either gel filtration or EM negative staining. The former assay should be realistic, since the authors have obtained sufficient amount of protein for crystallization. Furthermore, gel filtration can easily tell whehter chitin binding induces dimerization of Mg1LysM.

It is unfortunate that deletion of the N-terminal tail made Mg1LysM unstable. But the authors should test if point mutations (R2, D12, I and T) can disrupt the interaction between the two N-terminal tails. Data from this or a similar experiment are also important to support their conclusion.

Reviewer #3: (No Response)

**Part III – Minor Issues: Editorial and Data Presentation Modifications**

Reviewer #1: (No Response)

Reviewer #2: (No Response)

Reviewer #3: (No Response)

PLOS authors have the option to publish the peer review history of their article (what does this mean?). If published, this will include your full peer review and any attached files.

Reviewer #1: No

Reviewer #2: No

Reviewer #3: No
---

## [Decision Letter · Decision Letter 2]

25 May 2020

Dear Dr. Thomma,

We are pleased to inform you that your manuscript 'A secreted LysM effector protects fungal hyphae through chitin-dependent homodimer polymerization' has been provisionally accepted for publication in PLOS Pathogens.

As you can see that two reviewers were satisfied with your revised manuscript although reviewer #2 had some remaining concerns. We always consider all reviewers’ advice when making an editorial decision, and in this case, we understand that the requests from the reviewer #2 will require considerable time and effort. Considering that access to the lab is cumbersome due to the Covid-19 situation, and after discussion with members of the editorial board, we felt there was sufficient reviewer support to proceed with the manuscript. Nonetheless, reviewer #2 made reasonable suggestions that we encourage you to consider if this project will be pursued further.

Best regards,

Hui-Shan Guo

Associate Editor

PLOS Pathogens

Wenbo Ma

Section Editor

PLOS Pathogens

Kasturi Haldar

Editor-in-Chief

PLOS Pathogens

orcid.org/0000-0001-5065-158X

Michael Malim

Editor-in-Chief

PLOS Pathogens

orcid.org/0000-0002-7699-2064

Reviewer Comments (if any, and for reference):

Reviewer's Responses to Questions

**Part I - Summary**

Reviewer #2: (No Response)

**Part II – Major Issues: Key Experiments Required for Acceptance**

Reviewer #2: (No Response)

**Part III – Minor Issues: Editorial and Data Presentation Modifications**

Reviewer #2: (No Response)

PLOS authors have the option to publish the peer review history of their article (what does this mean?). If published, this will include your full peer review and any attached files.

Reviewer #2: No

---

## [Editor Report · Acceptance letter]

16 Jun 2020

Dear Dr. Thomma,

We are delighted to inform you that your manuscript, "A secreted LysM effector protects fungal hyphae through chitin-dependent homodimer polymerization," has been formally accepted for publication in PLOS Pathogens.

Best regards,

Kasturi Haldar

Editor-in-Chief

PLOS Pathogens

orcid.org/0000-0001-5065-158X

Michael Malim

Editor-in-Chief

PLOS Pathogens

orcid.org/0000-0002-7699-2064